# Identification of Unstable Ellagitannin Metabolites in the Leaves of *Quercus dentata* by Chemical Derivatization

**DOI:** 10.3390/molecules28031246

**Published:** 2023-01-27

**Authors:** Zhang-Bin Liu, Yosuke Matsuo, Yoshinori Saito, Yong-Lin Huang, Dian-Peng Li, Takashi Tanaka

**Affiliations:** 1Department of Natural Product Chemistry, Graduate School of Biomedical Sciences, Nagasaki University, 1-14 Bunkyo-machi, Nagasaki 852-8521, Japan; 2Guangxi Key Laboratory of Functional Phytochemicals Research and Utilization, Guangxi Institute of Botany, Guilin 541006, China

**Keywords:** vescalagin, liquidambin, ascorbic acid, phenylenediamine, oxidation, intermediate, ellagitannin, *Quercus dentata*

## Abstract

The identification of unstable metabolites of ellagitannins having ortho-quinone structures or reactive carbonyl groups is important to clarify the biosynthesis and degradation of ellagitannins. Our previous studies on the degradation of vescalagin, a major ellagitannin of oak young leaves, suggested that the initial step of the degradation is regioselective oxidation to generate a putative quinone intermediate. However, this intermediate has not been identified yet. In this study, young leaves of *Quercus dentata* were extracted with 80% acetonitrile containing 1,2-phenylenediamine to trap unstable ortho-quinone metabolites, and subsequent chromatographic separation afforded a phenazine derivative of the elusive quinone intermediate of vescalagin. In addition, phenylenediamine adducts of liquidambin and dehydroascorbic acid were obtained, which is significant because liquidambin is a possible biogenetic precursor of C-glycosidic ellagitannins and ascorbic acid participates in the production of another C-glycosidic ellagitannin in matured oak leaves.

## 1. Introduction

Ellagitannins are a group of hydrolyzable tannins having hexahydroxydiphenoyl (HHDP) esters connected mainly to glucose. Owing to their large structural diversity and rich chemistry stemming from the different location of the ester groups on the glucose core, oxidation of the HHDP aromatic rings, coupling with other metabolites, and oligomerization, ellagitannins attract considerable interest in organic and natural product chemistry [1,2,3,4,5]. Among the known ellagitannins, C-glycosidic ellagitannins form a distinct group represented by vescalagin (**1**) and its C-1 epimer castalagin (**2**) (Figure 1). These tannins are the major constituents of oak wood, used in barrel making [6,7,8]. During the aging of wine and whisky in oak barrels, tannins **1** and **2** degrade via autoxidation and react with coexisting compounds through chemical reactions that are important in the sensory evaluation of alcoholic beverages [9,10,11,12]. In this context, we previously investigated and proposed the chemical mechanisms involved in the autoxidation of oak ellagitannins and the subsequent addition of ethanol during whisky aging [12]. The related oxidative degradation of oak ellagitannin **1** is also observed in the growing young leaves of *Q. glauca* [13]. The degradation of **1** by a wood-rotting fungi (Shiitake mushroom, *Lentinula edodes*) in Japanese oak wood also proceeds according to the same mechanism [13]. These oxidative degradations, that is, autoxidation during the aging of spirits, the metabolism in oak young leaves, and degradation during mushroom cultivation, proceed via a common chemical mechanism, which is depicted in Figure 1. This mechanism was proposed on the basis of the isolation of intermediate **3** [13] or ethanol adduct **4** [12] having a cyclopentenone structure. The production of **3** in the young leaves of *Q. glauca* was confirmed by detection of an adduct **3a** after treatment with 1,2-phenylenediamine. In addition, when **1** was treated with mycelia of *Lentinula edodes*, **3** is accumulated in the mixture [13], and the structure of **3** was determined based on the spectroscopic data of **3a** obtained by treatment with 1,2-phenylenediamine. These results strongly suggested that the initial step of the degradation of **1** is the selective oxidation of the pyrogallol rings connected to the glucose C-1 position, in order to generate intermediate **1a** possessing a cyclohexenetrione structure. However, no direct chemical evidence has been provided to date for the generation of **1a** as the initial intermediate of the oxidation of **1**. Therefore, the aim of this study was to confirm the presence of **1a** and related unstable quinone metabolites in young oak leaves via the conversion of *ortho*-quinones to stable derivatives. In this study, 1,2-phenylenediamine was used as the derivatization reagent because it is conventionally used to trap diketone compounds in tannin chemistry. The leaves of *Q. dentata*, a deciduous tree widely distributed in Japan and East Asia, were selected because **1** and **2** are the major constituents of young leaves and their content decreases as the leaves grow. In addition, these leaves are very popular in Japan as a rice cake wrapper.

## 2. Results and Discussion

### 2.1. Comparison of Tannin Compositions in Spring and Summer Leaves

First, the tannin compositions in spring and summer leaves of *Q. dentata* were compared by high-performance liquid chromatography (HPLC) (Figure 2). Leaves collected in April and July were extracted with 80% acetonitrile (CH_3_CN) (samples A and C) and 80% CH_3_CN containing 1,2-phenylenediamine and trifluoroacetic acid (TFA) (samples B and D). The phenylenediamine used in the latter extraction conditions reacts with 1,2-diketone structures quantitatively to produce quinoxaline moieties under acidic conditions. As shown in Figure 2a, the major ellagitannins of the spring young leaves were **1** and **2**, whereas the content of both compounds decreased in the summer leaves (Figure 2c). In the HPLC of the summer leaves, broad peaks attributable to dimers and oligomers of **1** and **2** and grandinin (**9**) and related compounds [14,15,16], which have polyalcohol moieties at the C-1 position of **1**, were observed in the 6–12 min range, indicating that the content of oligomers and **9** and its analogues increases as the leaves grow. The HPLC of the extracts obtained with 1,2-phenylenediamine (Figure 2b, d) showed additional peaks attributable to derivatives produced via the reaction of 1,2-phenylenediamine, probably with metabolites having 1,2-diketone structures. Next, these derivatives were isolated in a larger-scale experiment.

### 2.2. Separation and Structure Determination

The fresh leaves collected at the end of April were extracted with 80% CH_3_CN containing 1,2-phenylenediamine and TFA, and separated by column chromatography using Sephadex LH-20, Diaion HP20SS, Chromatorex ODS, and Wakosil C18 columns to yield six compounds, including **1**, **2**, and three reaction products, i.e., **6**, **7**, and **8**. The unidentified peak **5** in the HPLC chromatograms shown in Figure 2 was identified as a dimeric ellagitannin cocciferin D_2_ (**5**) (Figure 3) [17] on the basis of the comparison of spectroscopic data, including two-dimensional (2D) nuclear magnetic resonance (NMR) experiments.

Compound **6** was isolated as a brown amorphous powder. The molecular formula was determined to be C_47_H_28_N_2_O_24_ according to the [M + H]^+^ peak at *m/z* 1005.1107 (calcd for C_47_H_29_N_2_O_24_, 1005.1110) and the [M + Na]^+^ peak at *m/z* 1027.0937 (calcd for C_47_H_28_N_2_O_24_Na, 1027.0910) observed in the high-resolution fast atom bombardment mass spectrometry (HR-FAB-MS) spectrum. The ^1^H and ^13^C NMR spectra (Table 1, Appendix A) showed signals ascribable to oxygenated aliphatic methine and methylene units, whose chemical shifts and coupling patterns were related to those of the glucose moiety of **1**. The ^1^H–^1^H correlation spectroscopy (COSY) confirmed the assignments of the glucose proton signals of **6**. The observation of two aromatic singlets at *δ*_H_ 6.61 (1H, s) and 6.88 (1H, s) and aromatic and carboxyl carbon signals showing heteronuclear multiple bond correlations (HMBC) (Figure 4) with the two aromatic protons revealed the presence of an HHDP group in the molecule. In addition, the HMBC correlations of the aromatic singlet signal at *δ*_H_ 6.76 (H_C_-6) and the similarities of the aromatic carbon signals with those of **1** indicated the presence of a nonahydroxytriphenoyl (NHTP) unit [6]. The location of the ester carboxyl carbons on the glucose core was determined according to the HMBC correlations. Specifically, the correlation of glc-H-6 (*δ*_H_ 4.04, 5.12) and HHDP-H_E_-6′ (*δ*_H_ 6.61) with HHDP-C_E_-COO (*δ*c 168.19) and that of glc-H-4 (*δ*_H_ 5.23) and HHDP-H_D_-6 (*δ*_H_ 6.88) with HHDP-C_D_-COO (*δ*_C_ 165.51) indicated that the HHDP group was connected to the 4- and 6-positions of glucose. The correlations of glc-H-5 (*δ*_H_ 5.72) and H_C_-6 (*δ*_H_ 6.76) to C_C_-COO (*δ*_C_ 165.92) suggested the presence of an ester linkage between glc-5 and the NHTP terminal galloyl unit. Furthermore, HMBC correlations between glc-H-2 (*δ*_H_ 5.47) and C_B_-COO (*δ*c 163.57) and C_A_-2 (*δ*c 116.13) and between glc-H-1 (*δ*_H_ 5.35) and C_A_-1, 2, 3 (*δ*c 135.0, 116.1, 151.8) were indicative of a C-glycosidic linkage of glc-1 to the NHTP ring A. These spectroscopic features are closely related to those of **1** and confirmed that **6** is derived from **1**. Meanwhile, the ^1^H and ^13^C NMR spectra of both compounds differ in the appearance of signals due to a phenylenediamine unit in the spectra of **6** (*δ*_H_ 8.22 (1H, br. d, *J* = 8.2 Hz), 8.11 (1H, br. d, *J* = 8.5 Hz), 7.93 (1H, m), 7.89 (1H, m); *δ*_c_ 141.63, 129.20, 131.8, 131.6, 129.6, 143.7) [13]. A comparison of ^13^C NMR chemical shifts of the aromatic carbons of **6** with those of **1** indicated that the ring A reacted with the phenylenediamine. The HMBC spectrum showed a correlation between glc-H-1 (*δ*_H_ 5.35) and the oxygen-bearing aromatic carbon signal at *δ*c 151.3 (NHTP-C_A_-3), indicating that the quinoxaline unit is located at the C_A_-4 and C_A_-5 positions of the NHTP moiety. The atropisomerism of the biphenyl linkages was the same as that of **1**, as was extracted from the comparison of the electronic circular dichroism (ECD) data with that of **1** [6]. Consequently, the structure of **6** was determined to be that shown in Figure 1, according to which the quinoxaline derivative **6** was generated from the missing intermediate **1a**, thus supporting the degradation mechanism of **1**, which was previously speculated from the structure of the degradation products [12,13].

The molecular formula of **7** was found to be C_47_H_32_N_2_O_25_ on the basis of the [M + H]^+^ peak at *m/z* 1025.1370 (calcd for C_47_H_33_N_2_O_25_, 1025.1372) observed in the HR-FAB-MS. Five aromatic singlets in the ^1^H NMR spectrum (Table 2, Appendix A) suggested the presence of a galloyl (*δ*_H_ 7.27 (2H, s, H_C_-2, 6)) and two HHDP (*δ*_H_ 6.84 (1H, s, H_B_-6), 6.73 (1H, s, H_D_-6), 6.44 (1H, s, H_E_-6), 6.40 (1H, s, H_A_-6)) groups in the molecule, which was confirmed by the corresponding ^13^C NMR, heteronuclear single-quantum correlation (HSQC), and HMBC spectroscopic analyses (Figure 4). The remaining aromatic proton signals at *δ*_H_ 7.69 (2H, dd, *J* = 6.0, 3.2 Hz, H_F_-4, 5) and 7.29 (2H, m, H_F_-3, 6)] were attributable to a benzene ring of the phenylenediamine unit. In addition, the ^1^H NMR spectrum exhibited six aliphatic proton signals due to four methines and one methylene of a polyalcohol moiety (*δ*_H_ 4.07 (1H, br. d, *J* = 13.2 Hz, H-6b), 4.98 (1H, dd, *J* = 13.2, 3.8 Hz, H-6a), 5.62 (1H, dd, *J* = 9.0, 1.3 Hz, H-4), 5.69 (1H, dd, *J* = 9.0, 3.8 Hz, H-5), 6.29 (1H, dd, *J* = 9.7, 1.3 Hz, H-3), and 6.35 (1H, br. d, *J* = 9.7 Hz, H-2)). The ^1^H–^1^H COSY correlations (Figure 4) and the coupling constants of the signals were related to those observed for **1**, suggesting the presence of a glucose moiety. The lack of a glucose C-1 methine proton was explained by the HMBC correlation of glucose H-2 with a *sp*^2^ quaternary carbon resonating at *δ*_C_ 146.7 (glc-C-1), suggesting that the C-1 atom of the polyalcohol is connected to the 1,2-phenylenediamine unit to form a benzimidazole unit. This is consistent with the molecular formula suggested by HR-FAB-MS. Two HHDP groups were determined to be connected at the polyalcohol 2,3- and 4,6-positions according to the HMBC correlations of the HHDP ester carbonyl carbons with the HHDP aromatic and polyalcohol aliphatic protons (Figure 4). Although the location of the galloyl group could not be determined on the basis of the usual HMBC experiment with *J* = 10 Hz, an HMBC spectrum measured with *J* = 5 Hz exhibited correlations between an ester carbonyl carbon (*δ*_C_ 165.1) and *δ*_H_ 5.69 (glc-H-5) and a two-proton singlet at 7.27 (galloyl H-2,6), confirming the location of the galloyl group at the C-5 position of the polyalcohol. These results suggested that the plane structure of **7** is that shown in Figure 4. This structure is related to liquidambin (**7a**), which has an open-chain form of the glucose core with an aldehyde group at the C-1 position (*δ*_C_ 194.6) [18]. Furthermore, the coupling constants of the polyalcohol moiety of **7** were very similar to those of the open-chain glucose unit of **7a** (*J*_1,2_ = 0 Hz, *J*_2,3_ = 9.5 Hz, *J*_3,4_ = 1 Hz, *J*_4,5_ = 9 Hz, *J*_5,6_ = 3.5 Hz, *J*_6,6_ = 13 Hz). The atropisomerism of the HHDP groups was also presumed to be the same as that of **7a,** because the close similarity of the coupling constants strongly suggests that **7** and **7a** adopted similar conformations including HHDP and galloyl esters. In addition, the HHDP unit located at the glucose 2,3- and 4,6-positions is known to adopt an *S*-configuration except for very limited examples [19,20]. This was confirmed by the ECD spectrum [21]. Consequently, **7** was concluded to be a reaction product of **7a** with 1,2-phenylenediamine, indicating the presence of **7a** in the young leaves of *Q. dentata*. Moreover, the structural similarity of **7a** with **1** and **2** suggests that **7a** is a biosynthetic precursor of the C-glycosidic ellagitannins of this plant [22]. So far, compound **7a** has been only isolated from *Liquidambar formosana*; therefore, the present study demonstrates the second identification of **7a** in plants.

The molecular formula of **8** was determined to be C_12_H_10_N_2_O_4_ on the basis of the FAB-MS (*m/z* 247 [M + H]^+^) and ^13^C NMR signals. The chemical shifts of the aromatic protons and carbons indicated the presence of a benzene ring of 1,2-phenylenediamine. The remaining six carbons are three *sp*^2^ carbons (*δ*c 165.9 (C-1), 140.4 (C-2), and 159.1 (C-3)), two methines (*δ*c 79.7 (C-4), 71.6 (C-5)), and a methylene (*δ*c 62.3 (C-6)). The HMBC correlations (Figure 5) between H-4 and the three *sp*^2^ carbons, including the lactone carbonyl carbon, suggested that **8** contained an unsaturated γ-lactone structure. In addition, the ^4^*J* HMBC correlation of H-4 with one of the aromatic carbon signals (*δ*c 143.7) indicated the condensation of 1,2-phenylenediamine with C-2 and C-3 to form a quinoxaline moiety. This is in agreement with the unsaturation index of 8 indicated by the molecular formula. These spectroscopic observations suggested that **8** is a reaction product of dehydroascorbic acid. This was confirmed by preparing **8** via the autoxidation of L-ascorbic acid in a neutral aqueous solution and subsequent condensation with 1,2-phenylenediamine. Thus, the structure of **8** was determined to be 3-(1,2-dihydroxyethyl)furo [3,4-b]quinoxaline-1-one [23]. The mechanism of formation of **8** is shown in Figure 2. A comparison of the young and matured leaves of *Q. dentata* by reversed phase HPLC (Figure 2) indicated an increase in the content of highly hydrophilic ellagitannins, which were detected as broad peaks with shorter retention times (t*_R_*, 6–12 min) compared with those of **1**. Previous studies of ellagitannins of *Quercus* species showed that these highly hydrophilic ellagitannins contain grandinin (**9**) and related oligomers, and ascorbic acid is suggested to participate in the biosynthesis of **9** [15].

## 3. Materials and Methods

### 3.1. General Information

Optical rotations were measured on a JASCO DIP-370 digital polarimeter (JASCO, Tokyo, Japan). Ultraviolet (UV) spectra were obtained on a JASCO V-560 UV/Vis spectrophotometer. Infrared (IR) spectra were measured on a JASCO FT/IR-410K IR spectrometer. ECD spectra were obtained using a JASCO J-725N spectrophotometer. ^1^H, ^13^C, and 2D NMR spectra were measured using a Varian Unity plus 500 spectrometer (500 MHz for ^1^H and 126 MHz for ^13^C, Agilent Technologies, Santa Clara, CA, USA) and a JEOL JNM-AL 400 spectrometer (400 MHz for ^1^H and 100 MHZ for ^13^C, JEOL Ltd., Tokyo, Japan). Mass spectra were obtained using a JMS-700N mass spectrometer (JEOL Ltd.). Column chromatography was performed using Sephadex LH-20 (25–100 mm, GE healthcare UK Ltd., Little Chalfont, UK), Diaion HP20SS (Mitsubishi Chemical Co., Tokyo, Japan), Chromatorex ODS (Fuji Silysia Chemical Ltd., Kasugai, Japan), Wakosil C18 (FUJIFILM Wako Chemicals Ltd., Osaka, Japan), and Toyopearl HW40F (Tosoh Corporation Ltd., Tokyo, Japan) columns. Thin-layer chromatography was performed on 0.25 mm-thick, precoated silica gel 60 F_254_ (Merck, Darmstadt, Germany) with toluene/ethyl formate/formic acid (1:7:1, *v*/*v*/*v*) and on 0.1 mm-thick, precoated cellulose F (Merck, Darmstadt, Germany) with 2% aqueous acetic acid. Spots were detected by illuminating with a short UV wavelength (254 nm) followed by spraying with 2% ethanolic FeCl_3_. Analytical HPLC was performed on a Cosmosil 5C18-ARII (Nacalai Tesque Inc., Kyoto, Japan) column (250 × 4.6 mm, i.d.) with a gradient elution of 40–30% (39 min) and 30–75% (15 min) of CH_3_CN in 50 mM H_3_PO_4_ at 35 °C and a flow rate of 0.8 mL/min. The HPLC system consisted of a JASCO PU-2080 Plus pump, a JASCO AS-2055Plus autosampler, a JASCO CO-2065Plus column oven, and a JASCO MD-2018 Plus photodiode array detector.

### 3.2. Plant Material

Leaves of *Q. dentata* were collected in April and July 2022 at the Bunkyo campus of Nagasaki University. A voucher specimen was deposited in the Graduate School of Biomedical Sciences, Nagasaki University.

### 3.3. HPLC Analysis

Fresh leaves (2.8 g) of *Q. dentata* collected at the end of April (samples A and B) and July (samples C and D) were homogenized using a Warring blender with 80% CH_3_CN (10 mL) for samples A and C and 80% CH_3_CN containing 1,2-phenylenediamine (250 mg/100 mL) and TFA (0.5%) for samples B and D. The homogenates were shaken in screw-capped vials at room temperature for 14 h. The extracts were passed through a membrane filter (0.45 um), and the filtrates were analyzed by HPLC.

### 3.4. Extraction and Separation

Fresh young leaves of sample B (56.1 g) were homogenized with 80% CH_3_CN (200 mL) containing 1,2-phenylenediamine (0.5 g) and TFA (2 mL). After extraction with shaking at room temperature for 14 h, the extract was filtered and the filtrate was concentrated to remove the organic solvent. After the removal of insoluble precipitates (mainly composed of chlorophylls) by filtration, the resulting aqueous solution was applied to a Sephadex LH-20 column chromatographer (3.0 cm i.d. × 22 cm) with a gradient elution of 0–100% MeOH (20% stepwise, each 300 mL) to give four fractions (Fr.1~4). Fr.2 (0.86 g); the solution was purified by Diaion HP20SS column chromatography (3.0 cm i.d. × 24 cm) with a gradient elution of 0–70% MeOH (10% stepwise, each 100 mL), Chromatorex ODS (2.0 cm i.d. × 24 cm) with an elution system of 0–100% MeOH (10% stepwise, each 100 mL), and Wakosil C18 (2.0 cm i.d. × 18 cm) with an elution system of 0–30% CH_3_CN (5% stepwise, each 100 mL) to give a mixture of **1** and **2** (163.5 mg) and **6** (48.9 mg). A similar separation of Fr.3 (0.25 g) afforded **7** (21.4 mg) and **8** (28.8 mg) together with a mixture of **1** and **2**. Compound **5** (156.7 mg) was isolated from Fr.4 (0.31 g) by Diaion HP20SS column chromatography.

#### 3.4.1. Compound **6**

Brown amorphous powder; [α${]}_{D}^{29.9}$−340.2 (c, 0.09, MeOH); UV (MeOH) λ_max_ nm (log ε): 379 (3.65), 2.87 (4.58), 257 (4.68), 234 (4.75), 214 (4.77); IR ν_max_ cm^−1^: 3356, 1737, 1713, 1613; ECD (MeOH) λ_max_ nm (Δε) 466 (−2.59), 380 (−2.83), 326 (+1.54), 301 (−1.61), 286 (+2.53), 261 (−41.35), 235 (+50.38), 217 (−6.79), 205 (+0.59); HR-FAB-MS *m/z* 1027.0963 [M + Na]^+^ (calcd for C_47_H_28_N_2_O_24_Na, 1027.0930), *m/z* 1005.1107 [M + H]^+^ (calcd for C_47_H_29_N_2_O_24_ 1005.1105); ^1^H and ^13^C NMR: Table 1.

#### 3.4.2. Compound **7**

Brown amorphous powder; [α${]}_{D}^{23}$ +53.1 (c 0.11, MeOH); UV (MeOH) λ_max_ nm (log ε): 278 (4.58), 212 (4.91), 205 (4.93); ECD (MeOH) λ_max_ nm (Δε) 311 (−1.90), 291 (+ 0.73), 263 (− 19.16), 236 (+ 52.20); IR ν_max_ cm^−1^: 3388, 1733, 1616, 1315, 1174; FAB-MS *m/z* 1025 [M + H]^+^; HR-FAB-MS *m/z* 1025.1370 [M + H]^+^ (calcd for C_47_H_33_N_2_O_25_ 1025.1367); ^1^H and ^13^C NMR: Table 2.

#### 3.4.3. (*R*)-3-((*S*)-1,2-Dihydroxyethyl)furo [3,4-b]quinoxaline-1(3*H*)-one (**8**)

Brown amorphous powder; [α${]}_{D}^{28}$ + 140.1 (c 0.09, MeOH); UV (MeOH) λ_max_ nm (log ε): 325 (3.83), 244 (4.38), 204 (4.40); IR ν_max_ cm^−1^: 3356, 1737, 1713, 1613. FAB-MS *m/z* 247 [M + H]^+^; ^1^H NMR (acetone-*d*_6_, 500 MHz) δ: 3.91 (1H, dd, *J* = 6.4, 10.7 Hz, H-6a), 3.93 (1H, dd, *J* = 7.8, 10.7 Hz, H-6b), 4.46 (1H, ddd, *J* = 1.5, 6.4, 7.8 Hz, H-5), 6.03 (1H, d, *J* = 1.5 Hz, H-4), 8.05, 8.10 (each 1H, m, H-4’, 5’), 8.28, 8.33 (each 1H, m, H-3’, 6’); ^13^C NMR (acetone-*d*_6_, 125 MHz) δ: 62.3 (C-6), 71.6 (C-5), 79.9 (C-4), 129.4, 130.8 (C-3’, 6’), 131.1, 133.1 (C-4’, 5’), 140.4 (C-2), 143.7, 143.9 (C-1’. 2’), 159.1 (C-3), 165.9 (C-1).

### 3.5. Preparation of 8 from L-ascorbic Acid

L-Ascorbic acid (10 mg) was dissolved in pH 7 citrate–phosphate buffer (2 mL) and stirred at room temperature for 18 h. The mixture was acidified to pH 3 with acetic acid, and 1,2-phenylenediamine (7 mg) was then added to the mixture. After stirring at room temperature for 12 h, the mixture was subjected to column chromatography using a Toyopearl HW40F (2 cm i.d. × 15 cm) with H_2_O containing increasing proportions of MeOH (0–100%) to give **8** (7.6 mg).

## 4. Conclusions

In this study, biosynthetic precursors **1a** and **7a** were isolated as derivatives **6** and **7**, respectively, via the reaction with 1,2-phenylenediamine. The isolation of **6** provides evidence for the presence of **1a**, which is a missing intermediate in the degradation of C-glycosidic ellagitannin. The presence of **7a** was also demonstrated by identifying **7**. A previous study suggested that **7a** is a precursor of C-glycosidic ellagitannins of *Liquidambar formosana* [21], and the structural similarity suggested that **7a** is also involved in the biosynthesis of **1** and **2** in the leaves of *Q. dentata*. Since diketone compound **3** exists mainly in its hydrated forms, triketone **1a** probably exists as a complex equilibrium mixture of hydrated forms in aqueous solutions [13], and **7a** was detected as a broad peak due to the equilibrium between the aldehyde and hydrated forms [18,21]. Therefore, the detection and isolation of intact **1a** and **7a** is difficult. Furthermore, a quinoxaline derivative of dehydroascorbic acid (**8**) was obtained. Since ascorbic acid is known to participate in the production of **9** from **1** in mature leaves [15], the observation of a large peak of **8** in the HPLC analysis of mature leaves (Figure 2) may be significant for the elucidation of the ellagitannin biosynthesis. To understand the biosynthesis and degradation of tannins and polyphenols, the characterization of ortho-quinones and related unstable intermediates is important; thus, this study demonstrated that extraction with 1,2-phenylenediamine and characterization of the resulting reaction products is a powerful method in polyphenol chemistry.

## Data Availability

Data is contained within the article or Appendix A.

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
