# Peer review of "Identification of Unstable Ellagitannin Metabolites in the Leaves of Quercus dentata by Chemical Derivatization"

_molecules, 2023, doi:10.3390/molecules28031246_

Round 1

Reviewer 1 Report

This paper deals with evidence for oxidated intermediate of C-glucosidic ellagitannin and proposes the mechanism of the oxidative degradation. The chemicals including NMR interpretation conducted appropriately I think, and the data would contribute further development of tannin chemistry.

Some minor points and queries are listed below.

・The numbering of HHDP and NHTP unit is not match to the previously reported papers [J Nat Prod, 78, 2104−2109, 2015]. Please check and revise if necessary.

・Calculated HR-FAB-MS value should be considered the weight of electron for comparison.

・Figure 2: Please insert the labels of solvent and season in the picture.

・Page 4, Line 124-126: Why this data can exclude the location of phenylenediamine unit adduction for other NHTP or HHDP moiety?

・Page 6, Line 150: The 1H-1H COSY correlations just indicate the presence of neighboring protons therefore authors should take other appropriate explanation for the presence of glucose moiety, or delete the sentence.

・Scheme 2: The details of reaction scheme for compound 9 should be depicted for enhance the readers good understanding. Also, the arrow for compound 9 would be started from dehydroascorbic acid, I think.

Author Response

・The numbering of HHDP and NHTP unit is not match to the previously reported papers [J Nat Prod, 78, 2104−2109, 2015]. Please check and revise if necessary.

There are two numbering system in ellagitannins: IUPAC system and a system based on ellagitannin biosynthesis. The numbering of this manuscript is based on consideration of biosynthetic pathway of ellagitannins. The HHDP, NHTP and other ellagitannin acyl groups are biosynthesized from gallic acid (3,4,5-trihydrozybenzoic acid). We think that this system is better in ellagitannin chemistry compared to IUPAC system.  

・Calculated HR-FAB-MS value should be considered the weight of electron for comparison.

Thank you for pointing it out. The calculated values was corrected.

・Figure 2: Please insert the labels of solvent and season in the picture.

According to the reviewer's advice, the Figure 2 was revised.

・Page 4, Line 124-126: Why this data can exclude the location of phenylenediamine unit adduction for other NHTP or HHDP moiety?

Thank you for pointing it out. Following sentence was added. "Comparison of 13C NMR chemical shifts of the aromatic carbons of 6 with those of 1 indicated that the ring A was reacted with the phenylenediamine."

・Page 6, Line 150: The 1H-1H COSY correlations just indicate the presence of neighboring protons therefore authors should take other appropriate explanation for the presence of glucose moiety, or delete the sentence.

The text was revised: The 1H–1H COSY correlations (Figure 3) and the coupling constants were related to those observed for 1, suggesting the presence of a glucose moiety.

・Scheme 2: The details of reaction scheme for compound 9 should be depicted for enhance the readers good understanding. Also, the arrow for compound 9 would be started from dehydroascorbic acid, I think.

The Scheme 2 was revised according to the advice.

Reviewer 2 Report

I think the manuscript is well wirtten and can be published with minor corrections:

1. More detailed information shoule be provided in the Introduction section.

2. Please check the references, for example, some references are with doi number while other not, why?

Author Response

  1. More detailed information shoule be provided in the Introduction section.

    According to the advice, explanation for production and identification of 3 was added in the introduction: "The production of 3 in the young leaves of Q. glauca was confirmed by detection of an adduct 3a after treatment with 1,2-phenylenediamine. In addition, when 1 was treated with mycelia of Lentinula edodes, 3 is first accumulated in the mixture [13], and the structure of 3 was determined based on the spectroscopic data of 3a obtained by treatment with 1,2-phenylenediamine."

In addition, the structure 3a was added in Figure 1.

  1. Please check the references, for example, some references are with doi number while other not, why?

Doi was added to the references, except for books.

Reviewer 3 Report

The work by Tanaka and co-workers describes the use of 1,2-phenylenediamine to trap intermediates/metabolites from Quercus dentata. The method proved to be useful and reliable since the authors were able to characterize a few metabolites The manuscript is well-written, it is sound, and the information/data presented is solid as evidence to demonstrate the usefulness of the technique, while validating the characterization. The work merits publication after two minor addition. 

1. add the name of the organic solvent in the abstract

2. Draw the structures of the molecules on the 1HNMR spectra. This will make it easy to follow and visualize/match the protons. 

Author Response

  1. add the name of the organic solvent in the abstract

   The organic solvent was revised to 80% CH3CN in Abstract.

  1. Draw the structures of the molecules on the 1HNMR spectra. This will make it easy to follow and visualize/match the protons. ã€€

   Structures were added in the spectra.